# Predictive Value of Reactogenicity for Anti-SARS-CoV-2 Antibody Response in mRNA-1273 Recipients: A Multicenter Prospective Cohort Study

**DOI:** 10.3390/vaccines11010120

**Published:** 2023-01-03

**Authors:** Min Joo Choi, Jung Yeon Heo, Yu Bin Seo, Young Kyung Yoon, Jang Wook Sohn, Ji Yun Noh, Hee Jin Cheong, Woo Joo Kim, Ju-yeon Choi, Young Jae Lee, Hye Won Lee, Sung Soon Kim, Byoungguk Kim, Joon Young Song

**Affiliations:** 1Department of Internal Medicine, International St. Mary’s Hospital, Catholic Kwandong University College of Medicine, Incheon 22711, Republic of Korea; 2Department of Infectious Diseases, Ajou University School of Medicine, Suwon 16499, Republic of Korea; 3Division of Infectious Diseases, Department of Internal Medicine, Kangnam Sacred Heart Hospital, Hallym University College of Medicine, Seoul 07441, Republic of Korea; 4Department of Internal Medicine, Korea University Anam Hospital, Korea University College of Medicine, Seoul 02841, Republic of Korea; 5Department of Internal Medicine, Korea University Guro Hospital, Korea University College of Medicine, Seoul 08308, Republic of Korea; 6Vaccine Innovation Center-KU Medicine (VIC-K), Seoul 08308, Republic of Korea; 7Division of Vaccine Clinical Research Center for Vaccine Research, National Institute of Infectious Diseases, Cheongju 28159, Republic of Korea

**Keywords:** COVID-19, SARS-CoV-2, vaccine, reactogenicity, immunogenicity

## Abstract

Messenger RNA (mRNA) vaccination was developed to mitigate the coronavirus disease 2019 pandemic. However, data on antibody kinetics and factors influencing these vaccines’ immunogenicity are limited. We conducted a prospective study on healthy young adults who received two doses of the mRNA-1273 vaccine at 28-day intervals. After each dose, adverse events were prospectively evaluated, and blood samples were collected. The correlation between humoral immune response and reactogenicity after vaccination was determined. In 177 participants (19–55 years), the geometric mean titers of anti-S IgG antibody were 178.07 and 4409.61 U/mL, while those of 50% neutralizing titers were 479.95 and 2851.67 U/mL four weeks after the first and second vaccine doses, respectively. Anti-S IgG antibody titers were not associated with local reactogenicity but were higher in participants who experienced systemic adverse events (headache and muscle pain). Antipyretic use was an independent predictive factor of a robust anti-SARS-CoV-2 antibody response after receiving both vaccine doses. Systemic reactogenicity after the first dose influenced antibody response after the second dose. In conclusion, mRNA-1273 induced a robust antibody response in healthy young adults. Antipyretic use did not decrease the anti-SARS-CoV-2 antibody response after mRNA-1273 vaccination.

## 1. Introduction

As the coronavirus disease 2019 (COVID-19) pandemic continues, vaccines against severe acute respiratory syndrome coronavirus 2 (SARS-CoV-2) have been rapidly developed to control the disease spread [1]. In South Korea, the national COVID-19 vaccination program has been gradually expanded to include the messenger RNA (mRNA)-1273 vaccine (Spikevax^®^, ModernaTX, Inc., Cambridge, UK), the fourth vaccine authorized as of May 2021 [2]. mRNA-1273 is a lipid nanoparticle-encapsulated mRNA vaccine that elicits an antibody response against the spike protein of SARS-CoV-2 and has shown favorable results in both clinical trials and real-world studies [3,4,5]. Nevertheless, data on the kinetics of antibodies after vaccination with mRNA-1273 are currently limited, particularly in Asian countries [6,7,8,9,10,11].

In clinical trials, mRNA-1273 proved more immunogenic and more likely to cause adverse events (AEs) than BNT162b2 [12]. Moreover, among those vaccinated with mRNA-1273, certain groups (e.g., younger population) showed a higher immune response accompanied by frequent AEs [6,13,14]. These synchronous associations may support the hypothesis that strong reactogenicity is associated with better immunogenicity. This correlation has been reported in previous studies, but the results have been inconsistent depending on the vaccine platform, vaccine type (antigen and adjuvant) and study population [11,15,16,17,18,19,20,21]. In this study, to shed light on these uncertainties, we evaluated the immunogenicity of mRNA-1273 and its correlation with AEs in healthy young adults.

## 2. Methods

### 2.1. Study Participants

This prospective cohort study was conducted in June 2021 at four university hospitals. Healthy young adults between the ages of 19 and 55 years, who were willing to receive the mRNA-1273 vaccine, were enrolled in the study; the participants provided written informed consent (Clinical Trial Number—NCT05258708). Individuals were excluded from the study if they were previously diagnosed with laboratory-confirmed COVID-19, had a history of autoimmune disease, or were immunocompromised, pregnant, or breastfeeding. Demographic information and data regarding the presence of comorbidities were collected from each participant. This study was approved by the institutional review board of each participating hospital.

### 2.2. Immunogenicity Assessment

Blood samples were collected at baseline (T0), 28 ± 7 days after the first dose but before the second dose (T1), and 28 ± 7 days after the second dose (T2). Two doses of mRNA-1273 were injected into the deltoid muscle of the upper arm at an interval of 28 days. To analyze immunogenicity, the anti-S antibody was measured using SARS-CoV-2 immunoglobulin G (IgG) (Elecsys anti-SARS-CoV-2 spike ECLIA, Roche Diagnostics, Pleasanton, CA, USA) according to the manufacturer’s protocol. For the analysis of factors influencing humoral immune response, the strong antibody response was defined as IgG antibody titers ≥250 U/mL (the threshold for further dilution) at T1 and ≥5400 U/mL (four-fold higher titer than that correlated with viral neutralization titer ≥160) at T2 [22]. The plaque reduction neutralization test was performed using the wild-type SARS-CoV-2 virus (BetaCoV/Korea/KCDC03/2020) [23]. The median neutralizing titer (ND_50_) was defined as the concentration of the antibodies that reduced the number of viruses by 50%, and a threshold ≥1:20 was considered positive.

### 2.3. Adverse Event Assessment

At 7 days after each vaccine dose, the participants were requested to record the occurrence, severity, and duration of solicited AEs through a standardized electronic questionnaire. Information on the use of antipyretics was collected after each vaccination dose. A standard scale was used to grade the severity of AEs [24]. The overall severity of AEs was evaluated in the following three ways: (1) the highest level of severity of the AEs reported by the subjects, (2) the sum of the severity scores (SUM) for each AE, and (3) the sum of multiplying each symptoms’ severity by the duration (days) of symptoms (SoM) for each AE.

### 2.4. Statistical Analysis

We analyzed the differences in AEs occurring after the first and second doses using McNemar’s and Wilcoxon signed-rank tests for paired dichotomous and continuous variables, respectively. A repeated-measures analysis of variance (ANOVA) was used to determine the changes in antibody titers by time points (T0–T2) within the group of participants. Log-transformed data were used to calculate the geometric mean titers (GMTs) with 95% confidence intervals (CI). Either the χ^2^ test or Fisher’s exact test was used for categorical variables, whereas Student’s *t*-test or one-way ANOVA was used to compare the continuous variables, followed by Scheffé’s test for multiple comparisons. Multivariate logistic regression analysis was performed to identify the factors predictive of a strong antibody response. For the correlation analysis, Spearman’s rank correlation coefficient was calculated. Statistical significance was set at *p* < 0.05. All statistical tests were performed using SPSS Statistics for Windows, version 24.0 (IBM Corp., Armonk, NY, USA).

## 3. Results

### 3.1. Baseline Characteristics of the Study Participants

A total of 179 adult volunteers who were scheduled to receive the two doses of the mRNA-1273 vaccine participated in this study (Figure 1). Blood samples were obtained from 171 (95.5%) participants at all three time points (T0, T1, and T2) and 177 (98.9%) at two time points (T0 and T1). The baseline demographics are summarized in Table 1. The mean age of the patients was 25 ± 3.8 (range, 20–55) years, and 70% of the participants were women. All participants were healthy with no comorbidities. The mean body mass index (BMI) was 21.5 ± 2.8 kg/m^2^.

The mean interval between vaccine dose 1 and dose 2 was 28.9 ± 2.3 (range 26–43) days. The mean intervals from dose 1 to follow-up time points were 23.7 ± 3.3 (20–42) days to T1 and 56.8 ± 1.8 (54–63) days to T2. The mean interval from dose 2 to T2 was 27.9 ± 3.0 (14–35) days.

### 3.2. Adverse Events

The AEs after each vaccine dose are summarized in Table 1 and Appendix A. After the first dose, nearly all participants (99.4%) reported at least one AE. The most common AE was pain at the injection site (96%), followed by the limitation of movement at the injection site (89.3%), muscle pain (73.4%), and fatigue (58.8%). Fever and chills were reported in > 20% of the participants. Most AEs were grade 1 or 2 in terms of intensity. However, 6.8% of participants reported grade 3 or higher fatigue and redness/swelling at the injection site, and 14.1% reported grade 3 or higher limitation of movement at the injection site. After the second dose, 98.2% of the participants reported at least one AE. The most common AE was pain at the injection site (94.2%), followed by limitation of movement (88.9%), muscle pain (82.5%), and fatigue (78.4%). More than 70% of participants reported fever, chills, and headaches. Overall, systemic AEs were more common and severe after the second dose than after the first dose of the vaccine. A total of five participants complained of grade 4 AEs (emergency room visits or hospitalizations). Among them, one patient developed the AEs after the first dose of the vaccine (fatigue, myalgia, and chills), whereas the other four patients developed the AEs after the second dose; one complained of vomiting, one complained of myalgia with chills, and the last complained of fatigue, myalgia, headache, and chills. All patients recovered without sequelae.

### 3.3. Antibody Immune Response after Vaccination

The GMTs of anti-S IgG antibodies at each time point are presented in Figure 2. The anti-S IgG antibody titer was 0.4 U/mL in all volunteers before vaccination. The GMTs were 178.07 U/mL (95% CI, 159.00–199.48) and 4409.61 U/mL (95% CI, 4082.25–4762.12) after the first and second doses, respectively, which showed a statistically significant increase over time (*p* < 0.001) (Figure 2A); the antibody titers at T1 showed a significant correlation with those at T2 (r = 0.547, *p* < 0.001).

Neutralizing antibody response was tested in 100 participants using the plaque reduction neutralization test. Before vaccination, the GMT of ND_50_ was 10.09 (95% CI, 9.13–11.16). At 4 weeks after the first dose (T1), GMT increased to 479.95 (95% CI, 394.37–583.98); 85% (85/100) of them showed ND_50_ titers exceeding 160. At 8 weeks after the first dose (4 weeks after the second dose; T2), all of them showed ND_50_ titers exceeding 160, with a GMT of 2851.67 (95% CI, 2481.99–3276.42). The neutralizing antibody titers increased significantly with the first and second doses from baseline to 8 weeks post-vaccination (*p* < 0.001; Figure 2B). Neutralizing antibody titers at T1 showed a significant positive correlation with those at T2 (r = 0.396, *p* < 0.001). The titers of anti-S IgG and neutralizing antibodies showed a strong positive correlation at each time point after vaccination (Appendix A).

### 3.4. Association between Antibody Response and Adverse Events

The temporal changes in antibody immune responses were compared between those with and without AEs up to 8 weeks after the first dose (Table 2). No significant difference was found in the IgG antibody titers between those with and without at least one AE (local or systemic) after the first dose. As for the individual AEs, participants with a headache after the first dose showed significantly higher antibody titers after 8 weeks than those without (*p* = 0.034). Although statistically insignificant, participants with a fever after the first dose had higher IgG antibody titers at 8 weeks post-vaccination (first dose) than those without (*p* = 0.056). A comparison of anti-S IgG antibody responses between those with and without any AEs after the second dose at T2 showed no significant difference in IgG antibody titers between them (local or systemic). Although statistically insignificant, participants with a fever (≥37.5 °C) showed higher IgG antibody titers than those without (*p* = 0.056).

Table 2 summarize the association between AEs after each dose of vaccine and neutralizing antibody titers. Participants with at least one AE (local or systemic) after the first dose did not show a significant difference in neutralizing antibody responses from those without AE. However, those with at least one systemic AE had higher neutralizing antibody titers over time (*p* = 0.06 at T2). Among individual AEs, only muscle pain after the first dose was significantly associated with higher neutralizing antibody levels at T2 (*p* = 0.018). After the second dose, participants with at least one AE (local or systemic) showed a neutralizing antibody response similar to those without. No significant difference was found in neutralizing antibody titers after the second dose with respect to individual AEs.

### 3.5. Association between Antibody Response and the Severity of Adverse Events

The correlation between AE severity and immunogenicity was evaluated using the following three parameters: the highest level of severity, SUM, and SoM. Overall, the highest level of severity of local or systemic AEs was not predictive of higher antibody titers in the short-term (<8 weeks) follow-up period (Appendix A). Similarly, no significant difference was found in the antibody responses according to the increase in SUM or SoM after any vaccine dose (Appendix A).

### 3.6. Association between Antibody Response and Antipyretic Use

Antipyretic use was more frequently observed after the second dose of the vaccine. A total of 107 (60.5%) and 155 (90.6%) participants reported the use of antipyretics after the first and second doses, respectively (Table 1). All of them took acetaminophen; 4 (3.7%) and 11 (7.1%) participants additionally took nonsteroidal anti-inflammatory drugs (NSAIDs) after the 1st and 2nd dose, respectively. Of the 100 participants with neutralizing antibody results, 60 (60%) and 94 (94%) took antipyretics after the first and second doses, respectively. At each time point, both IgG and neutralizing antibody titers were higher in the antipyretic group (Table 3). After the first dose, antipyretic users showed higher antibody titers than non-users over time (*p* = 0.022 for anti-S IgG antibody titers and *p* = 0.124 for neutralizing antibody titers at T2). After the second dose, anti-S IgG antibody titers were higher in antipyretic users at T2 than in non-users (*p* = 0.010); however, the results were statistically insignificant for neutralizing antibody titers (*p* = 0.753). Antipyretic use was significantly associated with systemic (*p* < 0.001 after dose 1 and *p* = 0.028 after dose 2) and local AEs (*p* = 0.060 after dose 1 and *p* = 0.019 after dose 2).

### 3.7. Multivariate Analysis

In 177 participants, four weeks after the first dose (T1), anti-S IgG antibody titers were 250 U/mL or higher in 43 patients (24.3%). A strong anti-S IgG antibody response (≥250 U/mL) was not significantly associated with age, sex, BMI, or individual AEs (Appendix A). At 4 weeks after the second dose (T2), the anti-S IgG antibody titer was ≥5400 U/mL in 60 patients among the 171 participants (35.1%). A strong antibody response (≥5400 U/mL) after the second dose was not associated with age, sex, or BMI. Among individual AEs, chills, and fever after the second dose was related to a strong antibody response (*p* = 0.051 and 0.063, respectively). Notably, antipyretic use after any vaccine dose was significantly associated with a strong antibody response at T2. In the multivariate analysis adjusted for age, sex, BMI, pain at the site of injection, and antipyretic use at each vaccine dose, antipyretic use at any vaccine dose was a significant predictive factor of the strong antibody response at T2 (Table 4).

## 4. Discussion

In the present study, we evaluated short-term humoral immune response in mRNA-1273 recipients up to 8 weeks after vaccination. The GMTs of anti-S IgG antibody were 178.07 and 4409.61 U/mL, and those of 50% neutralizing titers (ND_50_) were 479.95 and 2851.67 U/mL at 4 weeks after the first and second doses, respectively. Anti-SARS-CoV-2 antibody response after the second dose was positively correlated with systemic AEs (headache and muscle pain). Antipyretic use was an independent predictive factor for a strong antibody response after both the first and second doses.

Even low levels of neutralizing antibodies have been found to protect against SARS-CoV-2 [25]. However, the immune correlates of protection for antibody levels have not yet been established. Data from an efficacy trial of the ChAdOx1 nCoV-19 vaccine showed 50%–80% vaccine efficacy against symptomatic infections, with live virus neutralization titers of 68–247 [26]. The United States Food and Drug Administration (FDA) guideline for convalescent plasma initially recommended a target antibody titer of 160 [27]. In this study, the neutralizing antibody levels reached a titer level considered positive (≥160) based on the FDA recommendations in the majority of participants (85%) after only a single dose and all participants (100%) after the second dose. This finding highlighted robust immune response induction by the mRNA-1273 vaccine among young adult participants.

In this study, systemic AEs were more frequent after the second dose than after the first dose; more than 70% of the participants experienced fever, chills, headache, muscle pain, and fatigue after the second dose. These findings are consistent with the reports of a phase 3 clinical trial, but the frequency of AEs was higher in this study; this difference might be related to the study design and characteristics of the participants [3]. Assuming that antigenic priming of the immune system after the first vaccine dose might contribute to increased reactogenicity following the subsequent antigenic exposure [28], we hypothesize that increased reactogenicity after mRNA-1273 vaccination, especially the occurrence of systemic AEs after the second dose, is strongly associated with higher immunogenicity.

Vaccine antigens are recognized as potential pathogens by the pattern recognition receptors of the innate immune system, which results in the release of pyrogenic cytokines (interleukin [IL]-1, IL-6, tumor necrosis factor-α, and prostaglandin E2) and the subsequent cascade of immune responses [29]. Such post-vaccination immune responses may be accompanied by local or systemic AEs in vaccinated individuals. However, studies on the immunological correlates of reactogenicity in humans are limited, and they have reported inconsistent results depending on the vaccine type [11,15,16,17,18,19,20,21]. Both mRNA-1273 and BNT162b2 vaccines are the first human mRNA vaccines, which raised concerns about their immunological correlates with reactogenicity [3,30]. After the introduction of mRNA vaccines, several studies have investigated the correlation between immunogenicity and reactogenicity, with inconsistent results [11,18,20,28,31,32,33,34,35,36,37,38,39,40,41,42]. Some studies did not show a significant association, but those studies had limitations. In those studies, AEs were assessed as a total score or severity level, and more than half of the participants were elderly individuals (aged ≥ 80 years) [18,31,32,33,41]. Although the statistical significance of such an association was variable in other studies, systemic reactogenicity was related to a higher immune response, and the correlation was more prominent after the second dose [11,12,20,28,34,35,36,37,38,39,40,42,43]. Up to now, the association of mRNA-1273 vaccine immunogenicity with reactogenicity has been investigated in only three studies. Two of them included both kinds of mRNA vaccines (mRNA-1273 and BNT162b2) [12,40], while Coronavirus Efficacy (COVE) trial in the United States just focused on the mRNA-1273 vaccine [11]. According to the results of COVE trial, several systemic symptoms after the second dose were significantly associated with both the neutralizing and binding antibody titers at day 28 after the second dose. The lack of significant association between systemic AEs and neutralizing antibody titers in our study might be related to the limited number of cases. Nevertheless, the multivariate analysis of our study showed that anti-S IgG titer was significantly higher in antipyretics users at any vaccine dose. The antipyretics use would reflect the systemic reactogenicity, which might be associated with the immunogenicity of mRNA vaccine. In the situation that repeated mRNA vaccination would be anticipated, the findings in this study might provide a useful message that systemic reactogenicity might be a good sign of immune response, and it is not necessary to avoid taking antipyretics. However, most participants only took acetaminophen, so further research is needed to investigate the influence of NSAIDs.

Immunological correlation with reactogenicity can be explained as follows. (1) Similar to the mechanism proposed in the observational study of the influenza vaccine [44], which described a significant correlation between post-vaccination fever and immune response, a systemic reaction after vaccination may be an indicator of a healthy innate immune response. A systemic febrile response could be related to the activation of the innate immune system, which facilitates adaptive immune engagement and the subsequent antibody response [44,45]. (2) The inflammatory nature of lipid nanoparticles in mRNA vaccines, as a potential adjuvant, can be partially responsible for AEs and related to the intensity of eliciting protective immunity [46]. (3) The positive effect of the post-first-dose systemic reactogenicity on the antibody response after the second dose can reflect an association with memory B cell production. (4) Higher reactogenicity, particularly after the second dose, can be partially explained by the stronger anamnestic cytokine response with repeated vaccinations. To better understand and verify the observed immunological correlates with reactogenicity, it would be helpful to include the cytokines and other quantifiable factors in the future studies.

This study had some limitations. First, only short-term immune responses were investigated in mRNA-1273 recipients. A longitudinal follow-up should be planned to ensure that the relationship becomes clearer over time. Second, neutralization titers were measured in only half of the participants in our study; thus, the small number of patients could have yielded a statistically not significant correlation between AEs and neutralizing antibody responses. Finally, in this study, we focused on healthy young adults. Further studies, particularly in the elderly population, are required.

## 5. Conclusions

In conclusion, mRNA-1273 induced a robust humoral immune response in healthy young adults and the anti-SARS-CoV-2 antibody response was significantly stronger in participants who experienced systemic AEs and used antipyretics. Antipyretic use might be an objective indicator of systemic reactogenicity after vaccination. The use of antipyretics did not decrease the anti-SARS-CoV-2 antibody response after mRNA-1273 vaccination.

## Figures and Tables

**Figure 1 vaccines-11-00120-f001:**
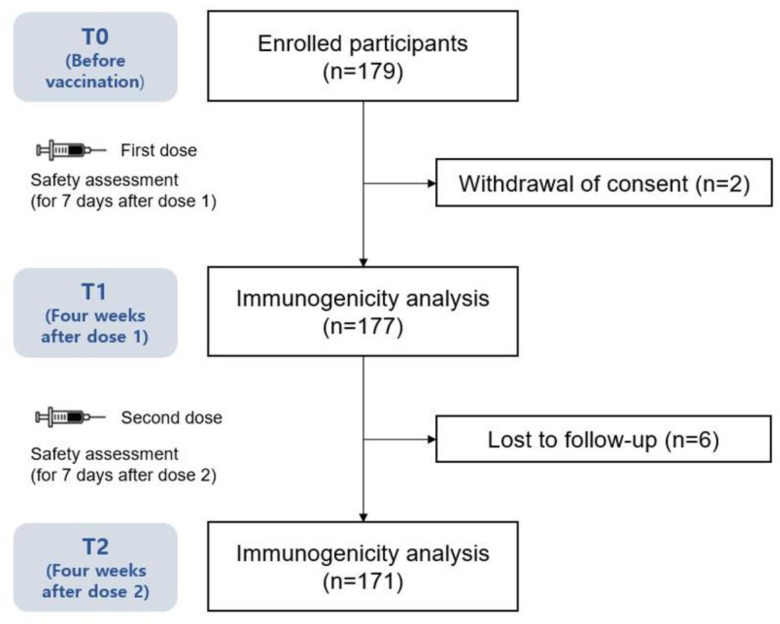
Study diagram.

**Figure 2 vaccines-11-00120-f002:**
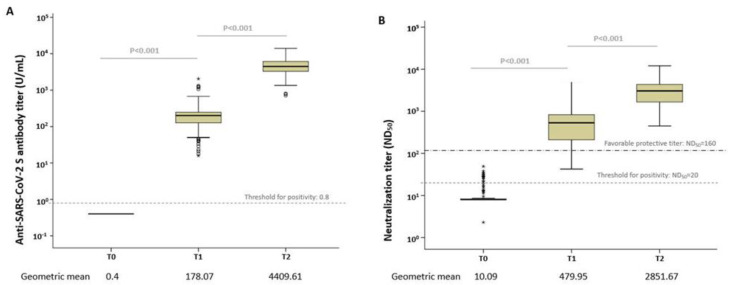
Box plots of the anti-SARS-CoV-2 antibody levels. (**A**) Anti-S IgG antibody and (**B**) median neutralizing titer (ND50) at each time point. The dotted line shows the threshold for positivity. Open circles depict outliers. Extreme values marked with a star.

**Table 1 vaccines-11-00120-t001:** Baseline characteristics and solicited adverse events after each vaccine dose.

Baseline Characteristics	Dose 1 (*n* = 177)	Dose 2 (*n* = 171)	*p*-Value
Age, mean ± SD (range)	25.4 ± 3.83 (20–55)	25.4 ± 3.88 (20–55)	
20–29 years	171	165	
30–39 years	4	4	
40–49 years	1	1	
50–54 years	1	1	
Male, number (%)	53 (29.9%)	53 (31.0%)	
BMI, mean ± SD	21.50 ± 2.87	21.58 ± 2.88	
**AE experienced after each dose**			
Any AE, number (%)	176 (99.4%)	168 (98.2%)	0.625
Any systemic AE, number (%)	155 (87.6%)	163 (95.3%)	0.007
fever	38 (21.5%)	125 (73.1%)	<0.001
chills	37 (20.9%)	121 (70.8%)	<0.001
headache	56 (31.6%)	120 (70.2%)	<0.001
muscle pain	130 (73.4%)	141 (82.5%)	0.038
fatigue	104 (58.8%)	134 (78.4%)	<0.001
joint pain	17 (9.6%)	45 (26.3%)	<0.001
vomiting	3 (1.7%)	16 (9.4%)	0.002
rash	13 (7.3%)	11 (6.4%)	0.815
dyspnea	5 (2.8%)	6 (3.5%)	1.000
flushing/lip swelling	5 (2.8%)	1 (0.6%)	0.219
facial palsy	5 (2.8%)	1 (0.6%)	0.219
paresthesia	15 (8.5%)	8 (4.7%)	0.189
Any local AE, number (%)	174 (98.3%)	164 (95.9%)	0.289
injection site pain	170 (96.0%)	161 (94.2%)	0.549
injection site redness/swelling	27 (15.3%)	38 (22.2%)	0.080
limited motion	158 (89.3%)	152 (88.9%)	1.000
**Severity of AE, mean ± SD ^a^**			
maximum severity	1.94 ± 0.75	2.29 ± 0.76	<0.001
highest grade of systemic AE	1.25 ± 0.77	1.98 ± 0.88	<0.001
highest grade of local AE	1.86 ± 0.74	1.87 ± 0.79	0.770
systemic SUM	2.99 ± 2.95	6.71 ± 4.06	<0.001
systemic SoM	4.71 ± 5.31	10.55 ± 8.00	<0.001
localized SUM	3.28 ± 1.53	3.40 ± 1.69	0.295
localized SoM	4.51 ± 2.34	4.74 ± 2.62	0.214
Antipyretic use, number (%)	107 (60.5%)	155 (90.6%)	<0.001

^a^ Severity was calculated by considering only some AEs (local AEs including pain, redness/swelling, motion limitation, and systemic AEs including fever, chills, headache, muscle pain, fatigue, joint pain, and vomiting). AE, adverse event; SUM, sum of each symptoms’ severity score; SoM, sum of multiplying each symptom severity by duration (days).

**Table 2 vaccines-11-00120-t002:** Relationship between reactogenicity and antibody response after the mRNA-1273 vaccine.

Type of AE	SARS-CoV-2 Antibody Assay		Participants with AEs after 1st Dose		*p*-Value	Participants with AEs after 2nd Dose		*p*-Value
			No	Yes		No	Yes	
Any local AE	Anti-S IgG (U/mL)	T1	N = 3	N = 174	0.332			
			274.92 (48.00–1575.07)	176.77 (157.65–198.20)				
		T2	N = 3	N = 168	0.394	N = 7	N = 164	0.658
			3435.58 (1395.40–8460.58)	4428.94 (4096.38–4788.51)		4054.15	4424.86 (4090.72–4787.40)	
	ND_50_	T1	N = 2	N = 98	0.66			
			652.38 (0.001–534195223)	476.87 (391.29–581.17)				
		T2	N = 2	N = 98	0.938	N = 1	N = 99	0.967
			2962.78 (24.46–358921.93)	2849.71 (2474.00–3282.46)		2934.95	2851.02 (2477.42–3434.79)	
Injection site pain	Anti-S IgG (U/mL)	T1	N = 7	N = 170	0.263			
			244.74 (145.64–411.24)	175.75 (156.42–197.51)				
		T2	N = 7	N = 164	0.765	N = 10	N = 161	0.836
			4166.77 (3113.15–5576-99)	4419.77 (4081.31–4787.40)		4267.76 (3067.61–5937.45)	4418.76 (4078.50–4786.30)	
	ND_50_	T1	N = 4	N = 96	0.373			
			310.89 (44.33–2184.74)	488.65 (400.50–596.21)				
		T2	N = 4	N = 96	0.957	N = 2	N = 98	0.912
			2798.98 (1629.67–4807.29)	2854.30 (2470.59–3296.86)		3011.62 (2171.20–4177.34)	2848.39 (2472.29–3282.46)	
Injection site redness/ swelling	Anti-S IgG (U/mL)	T1	N = 150	N = 27	0.206			
			181.97 (159.81–207.25)	157.91 (131.46–189.71)				
		T2	N = 144	N = 27	0.762	N = 133	N = 38	0.213
			4432.00 (4088.96–4807.29)	4290.42 (3377.54–5450.04)		4296.35 (3946.39–4676.27)	4830.59 (4018.83–5804.97)	
	ND_50_	T1	N = 80	N = 20	0.454			
			462.38 (367.45–581.70)	557.06 (382.65–810.96)				
		T2	N = 80	N = 20	0.664	N = 75	N = 25	0.153
			2808.67 (2386.16–3305.98)	3031.80 (2329.16–3947.30)		3022.04 (2569.21–3553.86)	2397.18 (1818.86–3158.64)	
Injection site motion limitation	Anti-S IgG (U/mL)	T1	N = 19	N = 158	0.758			
			187.46 (131.95–266.32)	177.01 (156.82–199.76)				
		T2	N = 19	N = 152	0.138	N = 19	N = 152	0.535
			3743.69 (3111.72–4504.02)	4500.91 (4140.00–4892.15)		4116.23 (3269.64–5183.22)	4447.34 (4095.43–4829.48)	
	ND_50_	T1	N = 8	N = 92	0.696			
			420.53 (154.06–1147.63)	485.51 (396.64–594.16)				
		T2	N = 8	N = 92	0.919	N = 4	N = 96	0.891
			2783.56 (1886.69–4105.82)	2857.59 (2462.63–3316.65)		2990.20 (659.63–13551.89)	2846.43 (2472.86–3276.42)	
Any systemic AE	Anti-S IgG (U/mL)	T1	N = 22	N = 155	0.769			
			186.29 (132.65–261.64)	176.97 (156.71–199.80)				
		T2	N = 22	N = 149	0.758	N = 8	N = 163	0.46
			4273.66 (3506.71–5208.35)	4429.96 (4071.93–4819.48)		3869.90 (2622.41–5710.84)	4438.13 (4099.21–4803.97)	
	ND_50_	T1	N = 12	N = 88	0.417			
			385.57 (175.15–848.79)	494.42 (403.92–605.34)				
		T2	N = 12	N = 88	0.06	N = 2	N = 98	0.912
			1998.02 (1256.90–3176.14)	2993.64 (2588.81–3461.78)		3011.62 (2171.20–4177.34)	2848.39 (2472.29–3282.46)	
Fever (≥37.5 °C)	Anti-S IgG (U/mL)	T1	N = 139	N = 38	0.764			
			179.72 (157.58–204.97)	172.31 (137.09–216.52)				
		T2	N = 136	N = 35	0.056	N = 46	N = 125	0.056
			4245.22 (3900.32–4621.68)	5108.58 (4259.91–6124.91)		3899.42 (3316.65–4584.58)	4613.18 (4228.63–5032.69)	
	ND_50_	T1	N = 79	N = 21	0.227			
			451.13 (360.08–565.20)	605.62 (400.50–916.01)				
		T2	N = 79	N = 21	0.866	N = 22	N = 78	0.842
			2834.00 (2414.90–3326.60)	2918.77 (2162.72–3939.13)		2777.15 (2071.57–3723.92)	2873.43 (2445.68–3375.20)	
Chills	Anti-S IgG (U/mL)	T1	N = 140	N = 37	0.309			
			183.53 (161.36–208.74)	158.93 (124.17–203.38)				
		T2	N = 137	N = 34	0.658	N = 50	N = 121	0.117
			4371.19 (4002.21–4775.29)	4565.62 (3885.08–5365.37)		4008.67 (3434.00–4679.51)	4586.70 (4197.59–5011.87)	
	ND_50_	T1	N = 79	N = 21	0.267			
			453.42 (362.41–567.28)	594.43 (387.61–911.59)				
		T2	N = 79	N = 21	0.698	N = 24	N = 76	0.731
			2811.90 (2407.13–3284.73)	3007.46 (2148.33–4210.17)		2731.49 (2128.63–3505.10)	2890.68 (2444.56–3419.01)	
Headache	Anti-S IgG (U/mL)	T1	N = 121	N = 56	0.266			
			170.49 (148.15–196.25)	195.66 (161.18–237.52)				
		T2	N = 117	N = 54	0.034	N = 51	N = 120	0.212
			4168.69 (3796.65–4577.20)	4979.66 (4353.11–5696.39)		4091.66 (3483.37–4805.07)	4551.98 (4173.50–4964.78)	
	ND_50_	T1	N = 72	N = 28	0.399			
			455.41 (358.53–581.70)	549.16 (396.10–761.20)				
		T2	N = 72	N = 28	0.399	N = 25	N = 75	0.99
			2747.89 (2329.7–3241.90)	3136.90 (2395.52–4106.77)		2847.08 (2099.91–3861.00)	2852.99 (2433.32–3345.80)	
Muscle pain	Anti-S IgG (U/mL)	T1	N = 47	N = 130	0.401			
			193.02 (160.10–232.65)	172.98 (150.42–198.93)				
		T2	N = 46	N = 125	0.676	N = 30	N = 141	0.958
			4308.24 (3837.96–4837.27)	4447.34 (4035.52–4901.17)		4425.88 (3809.78–5141.62)	4405.55 (4032.74–4813.93)	
	ND_50_	T1	N = 26	N = 74	0.162			
			379.75 (237.41–607.30)	521.07 (421.50–644.17)				
		T2	N = 26	N = 74	0.018	N = 9	N = 91	0.949
			2159.73 (1715.54–2718.94)	3144.13 (2661.95–3713.64)		2870.12 (2396.07–3438.75)	2849.71 (2447.37–3318.18)	
Fatigue	Anti-S IgG (U/mL)	T1	N = 73	N = 104	0.576			
			185.05 (157.80–217.02)	173.34 (147.84–203.24)				
		T2	N = 72	N = 99	0.593	N = 37	N = 134	0.885
			4519.60 (4017.91–5082.76)	4331.12 (3905.71–4803.97)		4362.14 (3634.96–5234.80)	4422.83 (4058.82–4818.37)	
	ND_50_	T1	N = 39	N = 61	0.902			
			472.61 (356.78–626.04)	484.62 (368.81–636.94)				
		T2	N = 39	N = 61	0.69	N = 18	N = 82	0.868
			2752.96 (2195.84–3452.23)	2916.76 (2433.88–3494.62)		2780.99 (1957.49–3951.85)	2867.48 (2456.97–3346.57)	
Joint pain	Anti-S IgG (U/mL)	T1	N = 160	N = 17	0.46			
			175.63 (155.52–198.34)	202.91 (147.84–278.48)				
		T2	N = 155	N = 16	0.298	N = 126	N = 45	0.481
			4352.11 (4015.13–4717.37)	5004.95 (3778.33–6629.79)		4337.11 (3948.21–4765.41)	4617.43 (4045.76–5271.08)	
	ND_50_	T1	N = 92	N = 8	0.352			
			466.98 (380.72–572.80)	657.05 (277.27–1557.04)				
		T2	N = 92	N = 8	0.886	N = 72	N = 28	0.639
			2843.15 (2456.97–3290.03)	2951.89 (1661.50–5243.24)		2793.83 (2365.92–3298.37)	3006.77 (2298.79–3932.78)	
Vomiting	Anti-S IgG (U/mL)	T1	N = 174	N = 3	0.267			
			179.60 (160.10–201.42)	109.42 (43.590 274.73)				
		T2	N = 168	N = 3	0.646	N = 155	N = 16	0.386
			4419.77 (4086.96–4779.69)	3853.90 (2535.71–5858.68)		4457.59 (4105.82–4839.49)	3967.35 (3168.11–4967.07)	
	ND_50_	T1	N = 98	N = 2	0.649			
			476.87 (390.30–582.51)	659.02 (97.54–4451.43)				
		T2	N = 98	N = 2	0.321	N = 90	N = 10	0.549
			2880.71 (2506.11–3310.55)	1748.24 (0.07–41409501)		2811.90 (2426.61–3258.37)	3236.68 (1976.97–5299.07)	
Rash	Anti-S IgG (U/mL)	T1	N = 164	N = 13	0.872			
			178.57 (158.60–201.00)	172.31 (110.71–268.16)				
		T2	N = 158	N = 13	0.688	N = 160	N = 11	0.548
			4428.94 (4099.21–4786.30)	4173.50 (2720.19–6404.72)		4382.28 (4045.76–4746.79)	4822.81 (3450.64–6742.17)	
	ND_50_	T1	N = 92	N = 8	0.16			
			460.57 (375.06–565.46)	770.19 (365.76–1622.18)				
		T2	N = 92	N = 8	0.558	N = 91	N = 9	0.516
			2886.69 (2494.59–3340.41)	2479.70 (1410.26–4360.14)		2893.34 (2490.00–3362.02)	2465.47 (1758.73–3455.41)	

Data are presented as geometric mean titers (95% confidence interval). AE, adverse event; SARS-CoV-2, severe acute respiratory syndrome coronavirus 2; IgG, immunoglobulin G; ND_50_, median neutralizing titer.

**Table 3 vaccines-11-00120-t003:** Relationship between the use of antipyretics and antibody response.

After Dose 1		Antipyretic Use		*p*-Value
		No	Yes	
Anti-S IgG (U/mL)	T1 (N = 177)	176.13 (148.56–208.82)	179.38 (153.93–209.03)	0.877
	T2 (N = 171)	3962.57 (3560.72–4409.77)	4748.31 (4269.28–5281.08)	0.022
ND_50_	T1 (N = 100)	411.50 (311.06–544.38)	531.72 (405.05–698.02)	0.206
	T2 (N = 100)	2498.72 (2028.41–3078.08)	3114.46 (2584.41–3753.21)	0.124
**After dose 2**				
Anti-S IgG (U/mL)	T2 (N = 171)	3234.86 (2427.75–4310.29)	4552.66 (4206.72–4927.06)	0.010
ND_50_	T2 (N = 100)	2611.77 (1803.82–3781.61)	2867.83 (2476.01–3321.66)	0.753
**After either dose**				
Anti-S IgG (U/mL)	T2 (N = 171)	2932.21 (2063.26–4167.13)	4547.28 (4207.95–4913.97)	0.004
ND_50_	T2 (N = 100)	2611.77 (1803.82–3781.61)	2867.83 (2476.01–3321.66)	0.753

Data are presented as geometric mean titers (95% confidence interval). IgG, immunoglobulin G; ND_50_, median neutralizing titer.

**Table 4 vaccines-11-00120-t004:** Multivariate analysis for predictive factors of strong antibody response four weeks after administration of the second dose of mRNA-1273 vaccine.

Variables	AEs after Dose 1		AEs after Dose 2	
	Odds Ratio, 95% CI	*p*-Value	Odds Ratio, 95% CI	*p*-Value
Age	1.009 (0.931–1.094)	0.822	1.023 (0.944–1.110)	0.577
Male	1.525 (0.674–3.449)	0.311	1.716 (0.718–4.102)	0.224
BMI	0.969 (0.854–1.100)	0.626	0.982 (0.863–1.116)	0.777
Injection site pain after dose 1	2.638 (0.297–23.399)	0.384		
Antipyretic use after dose 1	2.202 (1.110–4.367)	0.024		
Injection site pain after dose 2			0.511 (0.098–2.652)	0.424
Antipyretic use after dose 2			10.033 (1.185–84.924)	0.034

AE, adverse event; CI, confidence interval; BMI, body mass index.

## Data Availability

The data presented in this study are available within the article or Appendix A here.

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
