# Peer review of "Predictive Value of Reactogenicity for Anti-SARS-CoV-2 Antibody Response in mRNA-1273 Recipients: A Multicenter Prospective Cohort Study"

_vaccines, 2023, doi:10.3390/vaccines11010120_

Round 1

Reviewer 1 Report

The manuscript by Choi et al. described an interesting study to analyze the correlation between reactogenicity and vaccine immunogenicity in mRNA-1273 anti-SARS-CoV2 vaccine recipients. The study based on the hypothesis that strong reactogenicity is associated with better immunogenicity was conducted in 177 healthy adults with age ranging from 19 to 55, who received two doses of the mRNA vaccine in an interval of 28 days. Adverse event (AE, value of reactogenicity) assessment was recorded through participant questionnaire, while humoral immune responses, including anti-S antibody titer and ND50 neutralizing titer were measured with blood samples. The mRNA vaccine induced strong humoral immune response in this study. Despite that Most AEs are evident to associated with high antibody responses, some AEs such as headache, muscle pain showed some degree of significance in prediction. The authors further identified antipyretic use to be an independent predictive factor of a robust anti-SARS-CoV-2 antibody response after receiving both vaccine doses.

Despite the novelty and interesting hypothesis, most results did not give clear answer to the question that the authors tried to answer. This paper can be improved by addressing the following concerns.

Major:

1.     The age range of young adult (19-55) is too wide. Consider to group participants into multiple age groups

2.     Line 78-79 Blood sampling time point is critical. 28+/-14 days is too variable. Consider removing outliers and keep datapoints in narrow time point range.

3.     Severity of AEs could be subjective and difficult to measure and quantify. Future improvement of methods could include measurement of cytokines and other quantifiable factors.

4.     P value >0.05 is not statistically significant and should not be used to draw any conclusion.

5.     Line 38-39: “post-vaccination immunogenicity might be related to systemic reactogenicity.” This claim is very weak and should not be in the conclusion.

6.     Line 39-40: and 338-339 Result and analyses of this paper doesn’t lead to the conclusion that the use of antipyretics did not decrease the antibody titer. It only suggests that participants who experience severe AEs (and decide to take antipyretics) have higher antibody response. There is no direct and fair comparison for taking and not taking antipyretic in participants with similar systemic AEs.

Minor:

1.     Line37: misspell “conclusion”

2.     Please add some discussion of the potential application of this study, e.g. how to better predict vaccine efficacy based on self-assessment of AEs.

Author Response

Reviewer #1:

The manuscript by Choi et al. described an interesting study to analyze the correlation between reactogenicity and vaccine immunogenicity in mRNA-1273 anti-SARS-CoV2 vaccine recipients. The study based on the hypothesis that strong reactogenicity is associated with better immunogenicity was conducted in 177 healthy adults with age ranging from 19 to 55, who received two doses of the mRNA vaccine in an interval of 28 days. Adverse event (AE, value of reactogenicity) assessment was recorded through participant questionnaire, while humoral immune responses, including anti-S antibody titer and ND50 neutralizing titer were measured with blood samples. The mRNA vaccine induced strong humoral immune response in this study. Despite that Most AEs are evident to associated with high antibody responses, some AEs such as headache, muscle pain showed some degree of significance in prediction. The authors further identified antipyretic use to be an independent predictive factor of a robust anti-SARS-CoV-2 antibody response after receiving both vaccine doses.

Despite the novelty and interesting hypothesis, most results did not give clear answer to the question that the authors tried to answer. This paper can be improved by addressing the following concerns.

Major:

1.The age range of young adult (19-55) is too wide. Consider to group participants into multiple age groups

Answer) Thank you for advice. Mean age of participants was 25.4 +/- 3.8 years old; It was difficult to analyze by age group because 175/177 (98.9%) participants were between ages of 20-35 years, and only one was in their 40s and one was in their 50s.

We added age group distribution to table 1.

2.Line 78-79 Blood sampling time point is critical. 28+/-14 days is too variable. Consider removing outliers and keep datapoints in narrow time point range.

Answer) As you commented, we modified the sentence to 28 +/- 7 days, considering the outliers

Line 79-80: Blood samples were collected at baseline (T0), 28 ± 7 days after the first dose but before the second dose (T1), and 28 ± 7 days after the second dose (T2).

  1. Severity of AEs could be subjective and difficult to measure and quantify. Future improvement of methods could include measurement of cytokines and other quantifiable factors.

Answer) Thanks for the advice. The sentence was added to the discussion as follows.

Line 337-340: To better understand and verify the observed immunological correlates with reactogenicity, it would be helpful to include the cytokines and other quantifiable factors in the future studies.

4.P value >0.05 is not statistically significant and should not be used to draw any conclusion.

Answer) As you advised, fever was not statistically significant (p=0.056), we modified/removed the relevant conclusions.

Line 33-35: Anti-S IgG antibody titers were not associated with local reactogenicity but were higher in participants who experienced systemic adverse events (fever, headache and muscle pain).

Line 256-258: Anti-SARS-CoV-2 antibody response after second dose was positively correlated with several systemic AEs (fever, headache and muscle pain).

5.Line 38-39: “post-vaccination immunogenicity might be related to systemic reactogenicity.” This claim is very weak and should not be in the conclusion.

Answer) Thank you for advice. We remove that sentence from the abstract. 

6.Line 39-40: and 338-339 Result and analyses of this paper doesn’t lead to the conclusion that the use of antipyretics did not decrease the antibody titer. It only suggests that participants who experience severe AEs (and decide to take antipyretics) have higher antibody response. There is no direct and fair comparison for taking and not taking antipyretic in participants with similar systemic AEs.

Answer) Thanks for the advice. Individuals who took the antipyretic was more likely to have adverse reactions (as confirmed in our result, Line 233-235), so it is thought that the reactogenicity and the immunogenicity might be related.

However, as you said, it is difficult to compare the type and severity of AEs between those with and without antipyretics use, so we interpreted the results as conservatively as possible (taking antipyretics did not reduce immunogenicity). In this study, most participants only took acetaminophen; less than 10% took NSAIDs additionally. We added this point in the results and discussion. Nevertheless, this would be a meaningful conclusion in the clinical field for future vaccinations, so we added about future application in the discussion as follows.

Line 221-224: All of them took acetaminophen; 4 (3.7%) and 11 (7.1%) participants additionally took nonsteroidal anti-inflammatory drugs (NSAIDs) after the 1st and 2nd dose, respectively.

Line 317-322: In the situation that repeated mRNA vaccination would be anticipated, the findings in this study might provide a useful message that systemic reactogenicity might be a good sign of immune response, and it is not necessary to avoid taking antipyretics. However, most participants only took acetaminophen, so further research is needed to investigate the influence of NSAIDs.

Minor:

1.Line37: misspell “conclusion”

Answer) Thank you for detailed comment. We correct it.

2.Please add some discussion of the potential application of this study, e.g. how to better predict vaccine efficacy based on self-assessment of AEs.

Answer) Thanks for the advice. It is expected that our findings, along with recent evidence that person with systemic AEs would have higher immunogenicity, can be used as evidence that mitigate public anxiety about vaccine AEs and encourage not necessarily avoid taking antipyretics. We added that implication in the discussion as follows.

Line 317-322: In the situation that repeated mRNA vaccination would be anticipated, the findings in this study might provide a useful message that systemic reactogenicity might be a good sign of immune response, and it is not necessary to avoid taking antipyretics. However, most participants only took acetaminophen, so further research is needed to investigate the influence of NSAIDs.

Reviewer 2 Report

Choi et al have investigated 177 vaccinees for their antibody response to the COVID-19 mRNA vaccines, and the reactogenicity. After each dose serum antibody titers were assessed, and symptoms recorded. Anti-S IgG titers were not associated with higher local reactogenicity in vaccinees, but titers were higher in people who experienced fever, muscle pain and headache, systemic adverse events. Anti-fever treatments did not decrease the response to the second dose, but instead predicted a strong antibody response to the second dose. 

The manuscript is well written and easy to read. The many different variables, including age are thoroughly examined and evaluated.

A few misspellings, a missing reference, a missing Table (Table 5) are indicated on the manuscript.

Author Response

Reviewer #2:

Choi et al have investigated 177 vaccinees for their antibody response to the COVID-19 mRNA vaccines, and the reactogenicity. After each dose serum antibody titers were assessed, and symptoms recorded. Anti-S IgG titers were not associated with higher local reactogenicity in vaccinees, but titers were higher in people who experienced fever, muscle pain and headache, systemic adverse events. Anti-fever treatments did not decrease the response to the second dose, but instead predicted a strong antibody response to the second dose. 

The manuscript is well written and easy to read. The many different variables, including age are thoroughly examined and evaluated.

A few misspellings, a missing reference, a missing Table (Table 5) are indicated on the manuscript.

Answer) Thank you for detailed comment. We correct as follows.

- misseplling was corrected

- the reference was written in the next sentence, so we change the order

- table 4 was incorrectly named as table 5, so we correct it 

Reviewer 3 Report

1. The authors state in the introduction (lines 52-53) that there are few data regarding the kinetics of antibodies after vaccination with mRNA-1273, particularly in Asian countries. Authors should cite the work of Siangphoe et al (PMID: 36130187) regarding reactogenicity and immunogenicity with the mRNA-1273 vaccine.

2. The authors establish in the introduction (lines 58-61) the hypothesis that a strong reactogenicity is associated with a greater immunogenicity. They cite 4 studies that include 1 of the COVID19 vaccine (reference 17), 2 with HPV vaccines (references 14-15), and 1 with the hepatitis vaccine (reference 16). However, there are many studies with anti-COVID vaccines that address this hypothesis that the authors do not cite. For this reason, the authors should eliminate the citations referring to anti-HPV vaccines (references 14-15) and anti-hepatitis (reference 16), and replace them with others related to anti-COVID vaccines, for example that of reference 27, PMID: 35183384, PMID: 34960219, PMID: 36130187, PMID: 36016109, PMID: 35333697, doi.org/10.1101/2022.04.05.22273450, PMID: 35036028, PMID: 364 and 4591.8:945

3. In the fifth paragraph of the discussion, the authors should compare their results with those obtained in the studies of PMID: 35183384, PMID: 34960219, PMID: 36130187, PMID: 36016109, PMID: 35333697, doi.org/10.1101/2022.04. 05.22273450, PMID: 35036028, PMID: 36444543, and PMID: 35989281, which are related to reactogenicity and immunogenicity after vaccination against COVID19.

Author Response

Reviewer #3:

1.The authors state in the introduction (lines 52-53) that there are few data regarding the kinetics of antibodies after vaccination with mRNA-1273, particularly in Asian countries. Authors should cite the work of Siangphoeet al(PMID: 36130187) regarding reactogenicity and immunogenicity with the mRNA-1273 vaccine.

answer) Thank you for your advice. We added the work of Siangphoeet at al. to the citation.

Line 52-53: Nevertheless, data on the kinetics of antibodies after vaccination with mRNA-1273 are currently limited, particularly in Asian countries [6-11].

  1. Siangphoe, U.; Baden, L.R.; El Sahly, H.M.; Essink, B.; Ali, K.; Berman, G.; Tomassini, J.E.; Deng, W.; Pajon, R.; McPhee, R. Associations of Immunogenicity and Reactogenicity After Severe Acute Respiratory Syndrome Coronavirus 2 (SARS-CoV-2) mRNA-1273 Vaccine in the COVE and TeenCOVE Trials. Clinical Infectious Diseases 2022.

2.The authors establish in the introduction (lines 58-61) the hypothesis that a strong reactogenicity is associated with a greater immunogenicity. They cite 4 studies that include 1 of the COVID19 vaccine (reference 17), 2 with HPV vaccines (references 14-15), and 1 with the hepatitis vaccine (reference 16). However, there are many studies with anti-COVID vaccines that address this hypothesis that the authors do not cite. For this reason, the authors should eliminate the citations referring to anti-HPV vaccines (references 14-15) and anti-hepatitis (reference 16), and replace them with others related to anti-COVID vaccines, for example that of reference 27, PMID: 35183384, PMID: 34960219, PMID: 36130187, PMID: 36016109, PMID: 35333697, doi.org/10.1101/2022.04.05.22273450, PMID: 35036028, PMID: 364 and 4591.8:945

Answer) Thank you for detailed review. It is expected that the characteristics of each vaccine type might be substantially different, and the mRNA vaccine, new platform used for the first time in humans, is thought to need another consideration. We write that sentence to expresses that the association might differ depending on the vaccine platform, type, as well as the differences of the individual studies, so we included other vaccine type such as HAV and HPV.

We think that the meaning of that sentence was not clear, so we modified it as follows. In addition, we added the representative study for each formulation of COVID-19 vaccine to the citation.

Line 59-62: This correlation has been reported in previous studies, but the results have been inconsistent depending on the vaccine platform, vaccine type (antigen and adjuvant) and study population [11,15-21].

3.In the fifth paragraph of the discussion, the authors should compare their results with those obtained in the studies of PMID: 35183384, PMID: 34960219, PMID: 36130187, PMID: 36016109, PMID: 35333697, doi.org/10.1101/2022.04. 05.22273450, PMID: 35036028, PMID: 36444543, and PMID: 35989281, which are related to reactogenicity and immunogenicity after vaccination against COVID19.

Answer) Thank you for your detailed review and critical advice. All of the above study were added to citation. Critically, the work of Siangphoe et al., which was published online 20 September 2022, was not considered at the time of our writing. Some of sentences have been mofidied with the addition of the work of Siangphoe et al.

Since it is a comparison of the researches on the mRNA platform vaccine, we did not discuss the studies on Astrazeneca vaccine among the studies you recommended.

Line 294-317: After the introduction of mRNA vaccines, several studies have investigated the correlation between immunogenicity and reactogenicity, with inconsistent results [11,18,20,28,31-42]. Some studies did not show a significant association, but those studies had limitations. In those studies, AEs were assessed as a total score or severity level, and more than half of the participants were elderly individuals (aged ≥ 80 years) [18,31-33,41]. Although the statistical significance of such an association was variable in other studies, systemic reactogenicity was related to a higher immune response, and the correlation was more prominent after the second dose [11,12,20,28,34-40,42,43]. Up to now, the association of mRNA-1273 vaccine immunogenicity with reactogenicity has been investigated in only three studies. Two of them included both kinds of mRNA vaccines (mRNA-1273 and BNT162b2)[12,40], while Coronavirus Efficacy (COVE) trial in the United States just focused on the mRNA-1273 vaccine [11]. According to the results of COVE trial, several systemic symptoms after the second dose were significantly associated with both the neutralizing and binding antibody titers at day 28 after the second dose. The lack of significant association between systemic AEs and neutralizing antibody titers in our study might be related to the limited number of cases. Nevertheless, the multivariate analysis of our study showed that anti-S IgG titer was significantly higher in antipyretics users at any vaccine dose. The antipyretics use would reflect the systemic reactogenicity, which might be associated with the immunogenicity of mRNA vaccine.

  1. Siangphoe, U.; Baden, L.R.; El Sahly, H.M.; Essink, B.; Ali, K.; Berman, G.; Tomassini, J.E.; Deng, W.; Pajon, R.; McPhee, R. Associations of Immunogenicity and Reactogenicity After Severe Acute Respiratory Syndrome Coronavirus 2 (SARS-CoV-2) mRNA-1273 Vaccine in the COVE and TeenCOVE Trials. Clinical Infectious Diseases 2022.
  2. Levy, I.; Levin, E.G.; Olmer, L.; Regev-Yochay, G.; Agmon-Levin, N.; Wieder-Finesod, A.; Indenbaum, V.; Herzog, K.; Doolman, R.; Asraf, K. Correlation between Adverse Events and Antibody Titers among Healthcare Workers Vaccinated with BNT162b2 mRNA COVID-19 Vaccine. Vaccines 2022, 10, 1220.
  3. Yamamoto, S.; Fukunaga, A.; Tanaka, A.; Takeuchi, J.S.; Inoue, Y.; Kimura, M.; Maeda, K.; Ueda, G.; Mizoue, T.; Ujiie, M. Association between reactogenicity and SARS-CoV-2 antibodies after the second dose of the BNT162b2 COVID-19 vaccine. Vaccine 2022, 40, 1924-1927.
  4. Jubishi, D.; Okamoto, K.; Hamada, K.; Ishii, T.; Hashimoto, H.; Shinohara, T.; Yamashita, M.; Wakimoto, Y.; Otani, A.; Hisasue, N. The association between adverse reactions and immune response against SARS-CoV-2 spike protein after vaccination with BNT162b2 among healthcare workers in a single healthcare system: a prospective observational cohort study. Human Vaccines & Immunotherapeutics 2022, 1-10.
  5. Williams, E.; Kizhner, A.; Stark, V.S.; Nawab, A.; Muniz, D.D.; Tribin, F.E.; Carreño, J.M.; Bielak, D.; Singh, G.; Hoffer, M.E. Predictors for Reactogenicity and Humoral Immunity to SARS-CoV-2 Following Infection and mRNA Vaccination: A Regularized Mixed-Effects Modelling Approach. medRxiv 2022.
  6. Lim, S.Y.; Kim, J.Y.; Park, S.; Kwon, J.-S.; Park, J.Y.; Cha, H.H.; Suh, M.H.; Lee, H.J.; Lim, J.S.; Bae, S. Correlation between Reactogenicity and Immunogenicity after the ChAdOx1 nCoV-19 and BNT162b2 mRNA Vaccination. Immune Network 2021, 21.
  7. Takahashi, W.; Mizuno, T.; Hara, K.; Ara, Y.; Hurutani, R.; Agatsuma, T.; Fujimori, M. Association of Systemic Adverse Reactions and Serum SARS-CoV-2 Spike Protein Antibody Levels after Administration of BNT162b2 mRNA COVID-19 Vaccine. Internal Medicine 2022, 61, 3205-3210.

Round 2

Reviewer 1 Report

The authors have address my previous concern. I have no more comments.